# Individualizing Fluid Management in Patients with Acute Respiratory Distress Syndrome and with Reduced Lung Tissue Due to Surgery—A Narrative Review

**DOI:** 10.3390/jpm13030486

**Published:** 2023-03-08

**Authors:** Jan Benes, Jiri Kasperek, Olga Smekalova, Vaclav Tegl, Jakub Kletecka, Jan Zatloukal

**Affiliations:** 1Department of Anesthesiology and Intensive Care Medicine, Faculty of Medicine in Plzen, Charles University, 32300 Plzen, Czech Republic; 2Department of Anesthesiology and Intensive Care Medicine, University Hospital in Plzen, 32300 Plzeň, Czech Republic; 3Biomedical Centre, Faculty of Medicine in Plzen, Charles University, 32300 Plzen, Czech Republic; 4Fachkrankenhaus Coswig GmbH, Zentrum für Pneumologie, Allergologie, Beatmungsmedizin, Thoraxchirurgie, 01640 Coswig, Germany

**Keywords:** acute respiratory distress syndrome, thoracic surgery, fluids, fluid responsiveness, hemodynamics

## Abstract

Fluids are the cornerstone of therapy in all critically ill patients. During the last decades, we have made many steps to get fluid therapy personalized and based on individual needs. In patients with lung involvement—acute respiratory distress syndrome—finding the right amount of fluids after lung surgery may be extremely important because lung tissue is one of the most vulnerable to fluid accumulation. In the current narrative review, we focus on the actual perspectives of fluid therapy with the aim of showing the possibilities to tailor the treatment to a patient’s individual needs using fluid responsiveness parameters and other therapeutic modalities.

## 1. Vignette

A 63-year-old man with a history of arterial hypertension was admitted to the intensive care unit (ICU) for hypoxemic respiratory failure. For five days, he suffered a high fever and cough with progressive dyspnea. At admission, he was somnolent, extremely exhausted, and with an oxygen saturation of air at 68%. The chest X-rays that were performed showed bilateral opaque infiltrates, and he tested positive for SARS-CoV2. Because of progressive respiratory insufficiency coupled with decreased mentation, he was sedated, intubated, and admitted to the ICU. Under mechanical ventilation with intermediate positive end-expiratory pressure and increased inspiratory oxygen fraction, his saturation of arterial blood with oxygen improved, but his blood pressure dropped significantly. The treating physician was in doubt about whether to follow the sepsis-surviving recommendation of administering intravenous fluid resuscitation of 30 mL per kg body weight to reverse the possible hypovolemia or whether he should be rather restrictive in order to spare the lung tissue and start the infusion of norepinephrine. Therefore, he performed several examinations using ultrasonography of the vena cava and echocardiography of the heart. Initially, he used a concomitant small dose of vasopressors with 500 mL of Ringer’s lactate to reach an acceptable blood pressure target of mean arterial pressure of 65 mmHg. Afterwards he repeated the ultrasonography examination and performed several tests of fluid responsiveness too to titrate 250 mL fast infusions to reach the state of optimal preload and heart performance. However, the small dose of norepinephrine remained. The cumulative fluid balance of the patient during the first two days was 3.5 L positive, but his oxygenation as well as peripheral perfusion and also diuresis improved. This enabled the down-titration of the fluid infusion to enable a daily zero fluid balance, and after his lungs’ functions improved, the patient started weaning off the ventilation support. At that time, a small dose of diuretics was used to reach a negative fluid balance regardless of the need for a discrete dose of vasopressors. An echocardiographic examination demonstrated physiological variability of vena cava diameter and the patient was successfully extubated after the first spontaneous breathing trial.

## 2. Introduction

Since Latta’s first application of saline in the late 18th century, fluid therapy remains the cornerstone of our treatments in the ICU. However, the way we perform fluid management has evolved throughout these decades. It oscillated between liberal and conservative strategies, using artificial colloids or albumin, saline or balanced crystalloids. It took some years to recognize that fluids, as well as any other treatment, may induce harm either by changing the homeostasis [1], by affecting the blood coagulation [2], or by affecting the kidney functions [3]. Moreover, Lowell described the concept of fluid overload in 1990 [4], which evolved into fluid accumulation (any increase in weight induced by fluids) and fluid accumulation syndrome (FAS). The FAS has recently been defined as any amount of fluid accumulation that leads to organ dysfunction [5]. Based on clinical practice, the lungs are an organ extremely vulnerable to fluid accumulation, and multiple studies demonstrate the benefit of “dry” or “restrictive” regimes [6,7,8]. However, low circulating blood volume may lead to occult hypoperfusion affecting the kidney and other organs. Hence, finding the right balance is necessary concerning fluid therapy and management.

In patients with lung involvement, i.e., patients with acute respiratory distress syndrome (ARDS) or those after lung surgery, targeting appropriate fluid loading may significantly affect their outcome. Since the publication of Wiedemann’s FACCT trial [6], more data has been published on the faster liberation from ventilation in ARDS patients with restrictive strategies or the use of a proactive fluid removal [9]. ARDS is a syndrome coupled with increased alveolar-capillary permeability leading to interstitial and alveolar edema, which worsens with a positive cumulative balance [10,11]. This leads to a decreased gas exchange and affects the lungs’ elastic properties, propelling the vicious circle of the disease. In patients after lung resection surgery, the remaining tissue is affected by intraoperative trauma and inflammation. Moreover, it must take over the missing tissue and cope with deranged mechanical properties, gas distribution, and reduced vascular trees. Altogether, the patient with lung involvement needing fluid therapy poses a difficult task to manage. This narrative review summarizes the current knowledge about microcirculatory anatomy and physiology, the impact of endothelial glycocalyx, and up-to-date concepts of functional hemodynamic monitoring and fluid responsiveness to enable the personalization of fluid management.

## 3. Endothelium, Physiology, and Fluids

For decades, the body fluid compartmentalization between intra- and extravascular space has been described using the old paradigm of Starling [12]. Recent developments in our understanding of the endothelial barrier and endothelial glycocalyx (EG) have significantly changed this viewpoint [13,14]. The so-called revised Starling principle accents the role of different types of intercellular junctions between endothelial cells and the protein-rich surface of proteoglycans and integrated heparan or chondroitin sulphate molecules dividing the endothelial cells from the circulating plasma. This surface layer creates a highly effective barrier for the filtration of oncotic substances into the interstitium but also disables the hypothetical backward flow of fluids from the extravascular space at the end of the capillaries. Several authors have postulated that an extravascular fluid and protein leak is much more minor than previously thought, most of it returns into the circulation not by re-filtration on the venous side of the capillary but via the lymphatic flow [14]. Especially within the lung tissue resembling a sponge composed practically only from one layer of epithelial and endothelial cells separated by the basal membrane, the barrier functions of EG may play a crucial role.

However, EG is an extremely fragile structure. Its shedding exposes the endothelial cells with their ligand structures and enables an unopposed leak of the intravascular protein-rich plasma into the interstitium. The role of EG shedding in sepsis and viral infections has been demonstrated recently [15]. In addition, previous studies have shown that fluid overload and increased natriuretic peptides serum levels themselves may lead to EG shedding [16]. In patients undergoing surgical procedures, larger volumes of perioperative fluids increase the risk of lung complications in the postoperative course [8]. Neither EG-shedding molecules (heparan sulphate, etc.) nor natriuretic peptides have been analyzed in this retrospective study, so we can only hypothesize about the nature of such an association.

Moreover, the protective roles of EG are not limited to the fluid’s compartmentalization only. The proteoglycan endothelial surface layer serves as an effective mechanotransducer. This helps to cope with the capillary structure of the constant mechanical stress induced by the shear forces between the inside of the circulating blood-filled capillary and external mechanical strain caused by alveolar distension. In addition, this gelatinous layer impedes direct contact between plasma proteins or circulating blood elements and endothelium. EG-shedding exposes endothelium and subendothelial structures, enabling blood clotting via the interaction of plasmatic proteins and platelets with tissue factor, collagen, or von Willebrandt monomers [17]. It has been demonstrated that especially recent SARS-CoV2-induced viral lung damage consisted not only of damage induced to the alveolar cells but also to the endothelial part accompanied by intravascular thrombi formation [18]. Inflammation is another potent EG destructor, exposing the vessel wall to leukocytes and enabling their rolling and diapedesis [19]. Naturally, such interplay may severely affect the lung tissue, resulting in damage and functional failure such as ARDS.

## 4. ROSE Concept and Individual Fluid Management

When treating the critically ill, we can recognize the typical timing and aggressivity of the therapy, but not the source of the disease. For the time evolution of fluid treatment and accumulation, the R-O-S-E acronym (Figure 1) has been proposed [5]. Instead of Resuscitation, Optimization, Stabilization, and Evacuation, we can sometimes see other acronyms replacing the “R” with “S” as Salvage or “E” with “D” as De-escalation. Whatever the acronym, the descriptive nature of four phases of the critical illness is valid for most of our life-saving supportive treatments (cumulative fluid balance, dose of vasopressors, aggressivity of ventilation, and antimicrobial therapy); it is, therefore, pragmatic and easy to apply at the bedside.

## 5. Resuscitation

It has been demonstrated multiple times that initial aggressive care in patients with critical illness creates a potential for faster reversal. We have mentioned Latta’s salvage of cholera patients using saline infusion in the introduction. However, aggressive resuscitation only makes sense in those patients who are crashing. The difference between outcomes described by early goal-directed therapy in the famous Rivers study [20] and those demonstrated by later trials [21,22,23] may be (beside others) attributed to the fact that the mean oxygen saturation in the venous blood in the vena cava superior (ScvO_2_) of patients in the first trial were 47% and those in recent ones around 70%. From this point of view, the resuscitation phase is a salvage momentum at the last minute. Without a reasonable attempt to restore circulating blood volume, and/or managing his cardiovascular functions using drugs (mostly catecholamine), the patient is virtually doomed. The goal of resuscitation is the restoration of the minimal perfusion acceptable for the brain and heart, similar to the resuscitation of gas exchange by maintaining the airway patency and mechanically venting oxygen in and carbon dioxide out. However, the side effects of the treatments we perform at this moment are not irrelevant, especially when such an impact may be long-lasting. Massive fluid resuscitation with extravascular leak and edema formation may serve as an example. Multiple recent trials demonstrate lower cumulative fluid balance or better outcomes in those with a less aggressive initial resuscitation [24,25]. Therefore, the aggressivity should be rationally limited only to reach the desired effect. In moments of doubt, we should prefer treatments whose outcomes (both positive and negative) are short-lived (and hence reversible). From this perspective, fluids do have an extremely poor treatment profile. Usually, balanced crystalloids stay only a limited time in the intravascular space (though some context-sensitivity exists), and their volume expansion effect after about 20 min equals one-quarter of infused volume [26]. The remaining part diffuses into the interstitium and prompts edema formation. Using colloids may improve the volume effect, but only for a slightly longer time and at the cost of other (and in artificial colloids, more dangerous) side effects.

The treating physician frequently pays more attention to managing gas exchange in lung-compromised patients, but hemodynamic parameters cannot lie behind. Intubation and connection to positive pressure ventilation significantly change the conditions within the thorax and may result in hemodynamic compromise [27]. Typically, increased pulmonary hypertension induced by lung collapse and hypoxia-induced vasoconstriction results in right-sided heart overload. Administration of anesthesia and analgesia itself decreases the internal catecholamine surge. Still, these drugs have potential innate hemodynamic side-effects, including a negative inotropic effect, vasodilation leading to a decrease of venous return, and arterial vasodilation decreasing the tone and perfusion pressure. Based on published trials [27,28,29], several potential interventions are used to avoid post-intubation hemodynamic collapse. For instance, neither pre-induction volume loading nor pre-emptive catecholamines were associated with satisfactory results. Potentially because the postinduction collapse is multifactorial, and the proportion of divergent factors may differ between individuals. A practical approach speaks for adequate preparation with a stepwise increase of positive intrathoracic pressure via a non-invasive interface to avoid lung collapse, use a minimal dose of an intravenous anesthetic, and limit the use of drugs with adverse hemodynamic side effects (barbiturates, propofol). Catecholamines should be ready for administration—optimally run at a slow pace into an IV catheter to enable fast up-titration. By such a pragmatic approach, the use of fluid bolus may be unnecessary and temporarily replaceable by passive leg raising of the patient (if ventilator parameters allow). In some patients, rescue maneuvers may be necessary—recruitment or pronation. While the recruitment maneuver is accompanied mainly by hemodynamic side effects [30], prone positioning frequently positively impacts circulation by adapting ventilation-perfusion [31]. Before performing such rescue maneuvers, it may be wise to perform a fast bedside echocardiographic (ECHO) assessment to exclude hypovolemia, preferably on the right-side markers, i.e., diameter and collapsibility of the vena cava.

## 6. Optimization

Once the patient is out of the worst, further treatment should be extremely judicious. These steps aim to improve the whole body’s conditions and restore normal homeostasis and balance at the lowest cost of side effects. The patient’s background—i.e., comorbidities, nature of the disease, and time the critical illness developed—should be considered. In lung-compromised patients, this means careful titration of lung protective ventilation and using other rescue therapies first. From the cardiovascular perspective, we should try to reach the patient’s normal blood pressure values and reverse low cardiac output using a combination of fluids, vasopressors, and inotropes. Our aim should be the normalization of peripheral perfusion (absence of mottling, normalization of capillary refill) and reaching near-normal levels of serum lactate coupled with other laboratory markers of adequate cardiac output (ScvO_2_ and veno-atrial difference of carbon dioxide tension (P_va_CO_2_)). After protracted hypoperfusion, recovery of organ failure (elevated liver enzymes, dropped urine output, and increased serum creatinine) may be delayed and should not be regarded as imminent treatment failure.

While managing a patient with cardiovascular instability and lung impairment, it is recommended to use extended hemodynamic monitoring [32]. In actual clinical practice, bedside transthoracic echocardiography (TTE) and thermodilution techniques are the most frequently used. While TTE offers deep insight into the heart functionality, wall motion abnormalities, and valve pathologies, the thermodilution techniques enable operator-independent intermittent accurate scaling of the cardiac function. Some still propose a pulmonary artery catheter (PAC) in patients with severe ARDS to facilitate the assessment of the right ventricle function and monitoring of pressures within the pulmonary circulation. Nowadays, transpulmonary thermodilution (TPTD) is often preferred. The accuracy of the cardiac function flow parameters assessed using both thermodilution methods is comparable. Unlike PAC, the TPTD does not allow for monitoring of right-sided heart and pulmonary circulation, but TTE examination may help elucidate this information. On the contrary, TPTD enables a calculation of the global end-diastolic blood volume (GEDV, volumetric parameter of preload), extravascular lung water (EVLW, indicator of lung edema), and pulmonary permeability index (PVPI, indicator of endothelial permeability). Both EVLW and PVPI are sensitive markers and may help to guide fluid therapy much more precisely than filling pressures [33]. In addition, TPTD devices use arterial pressure curves for continuous monitoring of several hemodynamic parameters (first and foremost stroke volume) with high precision. Therefore, even small changes in cardiac function could be detected after diagnostic or therapeutic challenges in the cardiovascular system. This approach, called functional hemodynamic monitoring, enables the detection of preload responsiveness, guide fluid, and pharmacological therapy of shock states.

## 7. Testing Fluid Responsiveness

Because of the fluid accumulation risk, fluid responsiveness testing is highly recommended before any volume expansion [34]. This means using either existing heart-lung interactions (HLI) (i.e., during mechanical ventilation) or challenging the circulatory system (i.e., change of positive end-expiratory pressure (PEEP) level, performing end-expiratory hold, or giving a small volume challenge) to test whether the cardiovascular system will probably increase its performance after receiving real volume expansion. Nowadays, we have at hand multiple tests of preload responsiveness. Probably first in line is the variation of stroke volume (SVV) (or its proxy parameters—pulse pressure variation (PPV), plethysmography variability index (PVI), etc.). We are well aware of the usual limiting factors—regular heart rhythm without premature beats, absence of open thorax, and mandatory ventilation without spontaneous efforts and with a tidal volume of at least 8 mL/kg of ideal body weight [35]. These enable the use of ventilation-induced dynamic variations in most surgical patients. However, further limitations occur for the use in patients during and after lung surgery and in ARDS patients. The most important is the right ventricular failure combined with changes in the pulmonary vascular resistance [36]. These factors may lead to similar left ventricle stroke volume variations even though not correctable by fluid loading (i.e., false positive). In addition, high respiratory frequency and low heart rate interaction were also found to disqualify the method. Lung protective ventilation with small tidal volumes in patients with low compliance may lead to situations when the changes of intrathoracic pressures are not high enough to affect venous return, creating false negative findings. In such cases, a temporary increase of tidal volume from 6 to 8 mL/kg ideal body weight (so-called tidal volume challenge) may help to diagnose the positive fluid response (an absolute increase of PPV or SVV > 2.5% seems to be predictive by multiple studies [37]). Probably only a minority of patients have all the factors “in line” to enable relying on SVV or PPV. Moreover, in case of high SVV and/or PPV in patients with lung involvement, we should always actively look at the right side of the heart to exclude its failure.

Further tests may be used in routine: end-expiratory occlusion test (15 s lasting expiratory hold leading to increased stroke volume by 10% [38]). PEEP induced a decrease in flow parameters (14% decrease in stroke volume after the increase in PEEP by 5 cm H_2_O [39]). Some degree of fluid responsiveness may be diagnosed by performing lung recruitment—a 30% drop of stroke volume after 30 s lasting 30 cm H_2_O recruitment maneuver [40]. The latter seems to work even in a prone position [41], which is frequently used in ARDS patients and limits performing other tests. Sonographic assessment of inferior vena cava (IVC) diameter (and its respiratory-induced variation) is very easy to achieve. However, in patients with a risk of right ventricular dysfunction, the IVC may be engorged. Hence, only a small diameter with high variability indicates a positive preload response.

All previously listed tests are dependent on the HLI. Therefore, they may be affected by right ventricle dysfunction and pulmonary hypertension. Under such conditions, left ventricle preload may be limited, but fluid infusion will not improve cardiac output. Vice versa, it may increase right-sided congestion—observable by an increased central venous pressure (CVP) or IVC diameter. A real volume challenge may be necessary as a second step in case HLI tests are positive. This may be done using passive leg raising (PLR—a reversible mobilization of approximately 300 mL from abdominal and leg unstressed venous volume) or via fluid infusion. Performing passive leg raising is not without limitations [42], but clinical routine offers the best risk/benefit ratio. In situations when PLR is limited, and a high risk of lung edema is present (high EVLW, high PVPI), one may use the mini-fluid challenge test—rapid bolus of 60–100 mL via a central venous catheter with simultaneous observation of VTI of stroke volume with the continuous method. A slight increase (5–6%) of the screened flow parameter indicates a positive fluid response [43,44]. Finally, a 250 to 500 mL volume challenge may be given as a test treatment. Close observation of left-sided flow parameters and right-sided pressures should be performed in such a situation. Any sudden increase in CVP should prompt a stop of expansion and reassessment of the cardiovascular system [45]. The Figure 2 summarizes the simplified algorithm of the resuscitation and optimization phases.

## 8. Using Pharmacotherapy to Limit Fluids

To some extent, vasoactive medication may further decrease the need for fluids. Especially in the vasodilatory shock state, norepinephrine has been demonstrated to reduce the number of fluids administered to control the shock state [46]. Vasopressin seems similar to the volume expansion effect of concomitantly administered fluids [47] and may further improve hemodynamic stability and organ functions [48,49]. In patients with lung involvement, vasopressin improves systemic circulation seemingly without adverse effects on the pulmonary vasculature. Based on experimental data, unlike norepinephrine and another catecholamine, the V1R receptors are sparse in pulmonary circulation; hence vasopressin administration does not increase pulmonary resistance, at least based on experimental data [50,51,52] and human case report [53]. Adding vasopressin to norepinephrine early on in patients with pulmonary involvement may be a reasonable step to decrease fluid needs, better preserve organ perfusion, and stabilize the patient faster. However, this hypothesis has not been tested in real life yet.

## 9. Stabilization and De-Escalation

The transition between optimization and stabilization sometimes seems slightly blurred. The best marker of patients’ stabilization is the absence of a need to escalate vital organ support (ventilation, circulation). On the other hand, support of some other organs (i.e., renal replacement) may be necessary throughout this phase due to initial failure. The stabilization process may last days or weeks, and outbursts of complicating conditions may require repeated titration of ventilator setting and cardiovascular supports. Still, the overall picture is of a patient who is not significantly worsening further on. From day two to three, fluids given to optimize preload are not the primary driver of cumulative fluid balance. On the contrary, maintenance and dilution account for most fluid intake [54]. Based on the conditions, dilution of antimicrobials, sedation, and all other drugs frequently equals 2 L, and nutrition could make another one to two. For this reason, maintenance fluids (i.e., fluids given to replace patients’ inability to drink) are primarily unnecessary and should be carefully titrated [55]. During the stabilization phase, the patient’s daily fluid balance (i.e., all fluids intake minus diuresis and other fluid losses) is optimally neutral. However, the administration of diuretics may be necessary to reach this in patients with vasoactive substances and positive pressure ventilation. In the cornerstone FACCT trial [6], the conservative strategy based on early diuretics administration to maintain lower filling pressures despite low vasopressor or inotrope dose was coupled with significantly lower fluid balance, improved pulmonary function, and shorter mechanical ventilation. Similar results were replicated in the FACCT Lite trial with a simplified design [7].

The FACTT protocols demonstrate that reaching zero fluid balance during stabilization is favorable. However, in some patients, diuretics cannot maintain fluid balance. Oliguria due to acute kidney injury after a shock state is frequent among the critically ill [56]. In optimally loaded patients, a furosemide stress test (1–2 mg/kg intravenous furosemide bolus) may help to distinguish those with oliguric and non-oliguric renal injury [57]. In patients with non-oliguric injury, a following diuretic therapy may be used as in FACTT trials to reach zero balance. However, a rising cumulative fluid balance often occurs in those with oliguric injury, and renal replacement with active ultrafiltration is necessary. In patients with ongoing sepsis/septic shock, several studies have demonstrated the benefit of a “wait and see” strategy regarding the initiation of renal replacement [58], but in those with lung involvement and ARDS, letting the cumulative balance rise further after day three may be detrimental.

## 10. Active Evacuation

Usually, the disease gets under control after several days and vasoactive medication may be slowly weaned off. Some patients start to mobilize the retained water and spontaneously increase diuresis, but in most of them, active evacuation speeds up this process. This may be necessary, especially in patients with impaired gas exchange (i.e., ARDS) and those with signs of fluid accumulation syndrome (FAS). Venous excess ultrasound score (VExUS) has been proposed recently to help to monitor the risk of FAS [59], and VExUS congestion grade two or three should prompt active evacuation (though not supported by solid evidence) [60]. To reach a negative balance, several authors proposed the so-called PAL treatment [61], consisting of a high PEEP approach alleviating pulmonary edema, concentrated albumin administration to mobilize interstitial fluids, and diuretic therapy or extracorporeal ultrafiltration to reach a negative balance. In several trials [61,62], such an approach was associated with improved respiratory outcomes. However, the evacuation process should not be deliberate, and we must monitor the patient’s hemodynamic status and laboratory values (sodium, osmolality, and blood urea nitrogen) very carefully. Sometimes we speak about a late goal-directed fluid removal strategy [5]. Our aim should be on the physiologic level of fluid responsiveness without signs of low cardiac output and standard laboratory. In some patients, the evacuation is not without the need to return to a small dose of vasopressors, especially during intermittent sedation periods needed to overcome disturbing and/or uncomfortable moments of intensive care.

## 11. Future Perspectives and Conclusions

Individualizing fluid therapy in patients with ARDS or other lung involvement and impaired gas exchange is challenging. Maintaining the notion that fluids are drugs and should be used with caution and only in doses absolutely needed is critical for the success of the treatment. Up to date, we do not have a specific monitoring tool to assess the fluid loading conditions and amount of accumulated water. Currently, our possibilities of monitoring are expanding, having bioimpedance monitoring tools to evaluate the body water composition [63,64] and improving hemodynamic monitoring to assess cardiovascular functions and fluid responsiveness. Nevertheless, we lack a specific treatment for the globally increased permeability syndrome and endothelial glycocalyx disruption. Moreover, the fluids used to treat low circulating blood volume are far from ideal, and our research activities should focus on these points. For our routine clinical practice, the R-O-S-E concept, with its triggering parameters, preload responsiveness assessment, and individual goals, is the optimal approach to fluid stewardship in the ICU, especially for patients with ARDS or other lung involvement.

## Figures and Tables

**Figure 1 jpm-13-00486-f001:**
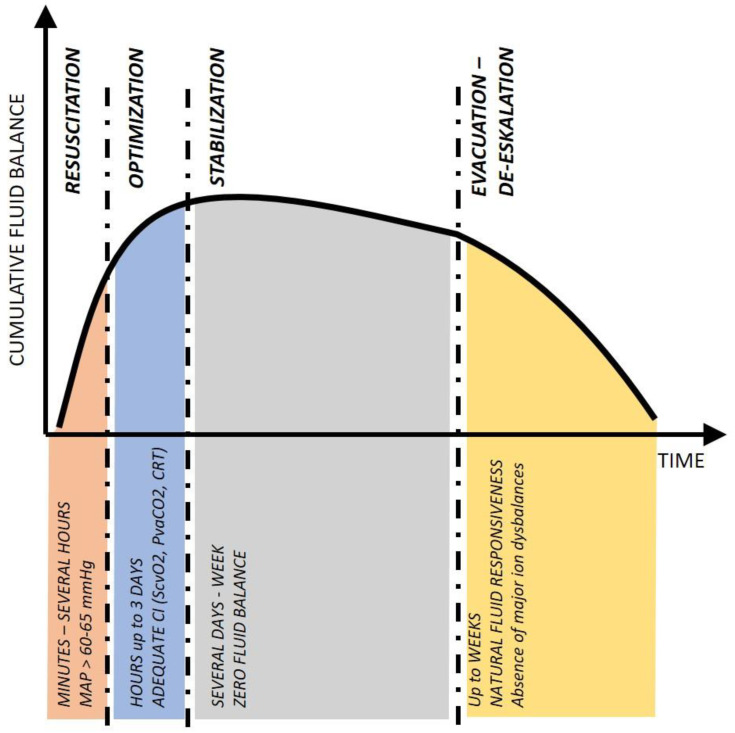
The ROSE concept of fluid therapy with the timing of each phase and potential treatment goals. *Abbreviations: CI—cardiac index, CRT—capillary refill time, MAP—mean arterial pressure, P_va_CO_2_—veno-arterial difference of CO_2_, S_cv_O_2_—central venous oxygen saturation*.

**Figure 2 jpm-13-00486-f002:**
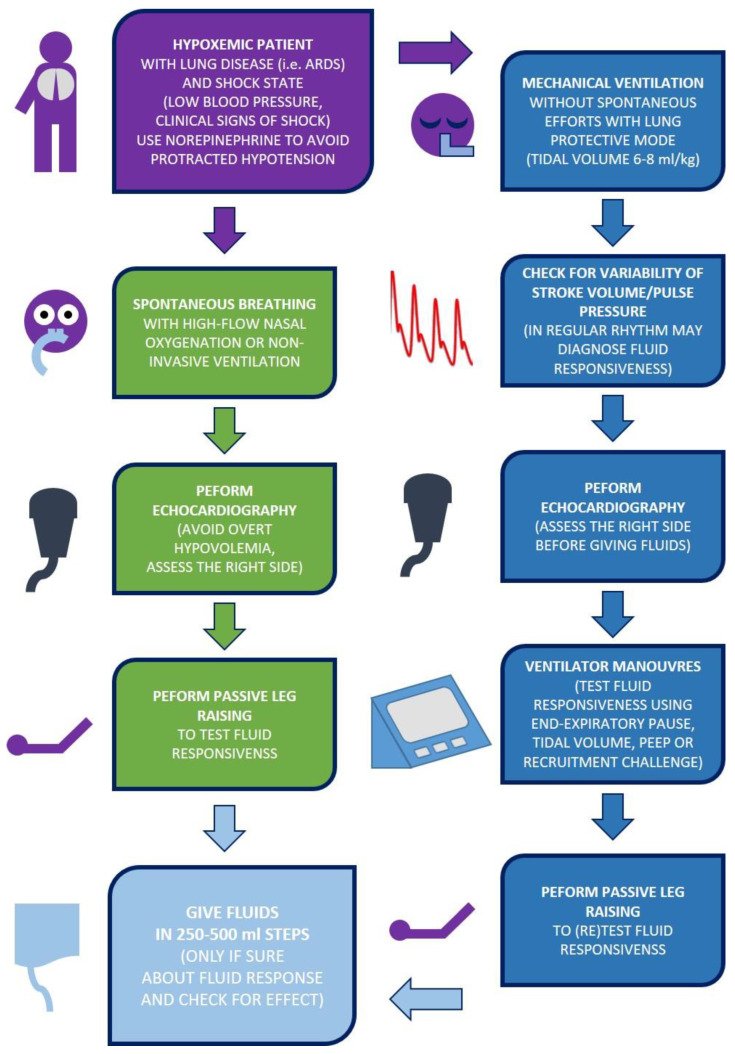
The simplified schematic approach to fluid therapy in patients with lung involvement during the resuscitation and optimization phase.

## Data Availability

Not applicable.

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
