# Peer review of "Individualizing Fluid Management in Patients with Acute Respiratory Distress Syndrome and with Reduced Lung Tissue Due to Surgery—A Narrative Review"

_jpm, 2023, doi:10.3390/jpm13030486_

Round 1

Reviewer 1 Report

Dear Authors,

thank you for presenting a very interesting and comprehensive review on fluid management in critcally ill pattients. The paper is well organized and the shows authors' expertise in the field. I have only minor remarks:

1. Not all abbreviations are clarified (e.g. ARDS, ICU)

2. Some short clinical vignette showing the practical use of presented knowledge would increase the value of the manuscript

Author Response

Reviewer 1

Dear Authors,

thank you for presenting a very interesting and comprehensive review on fluid management in critcally ill pattients. The paper is well organized and the shows authors' expertise in the field. I have only minor remarks:

  1. Not all abbreviations are clarified (e.g. ARDS, ICU)

Thank you for this proposal, which was also mentioned by the second reviewer. We have adopted the manuscript accordingly.

  1. Some short clinical vignette showing the practical use of presented knowledge would increase the value of the manuscript

Reviewer 2 Report

t of patients The article described for clinicians the difficult management of ARDS patients. Even though we did not find new strategies, the article presented well the main principles of treatment. With enough new articles from recent years, the article needs only minor revision. The authors must add the legend with the abbreviations appearing in the text for doctors with other specialties than intensive care.

Maybe an algorithm could help practicians better.  

Author Response

Reviewer 2

t of patients The article described for clinicians the difficult management of ARDS patients. Even though we did not find new strategies, the article presented well the main principles of treatment. With enough new articles from recent years, the article needs only minor revision. The authors must add the legend with the abbreviations appearing in the text for doctors with other specialties than intensive care.

Thank you for this proposal, which was also mentioned by the second reviewer. We have adopted the manuscript accordingly.

Maybe an algorithm could help practicians better.  

We have added a figure 2 displaying the major points of the approach in scheme
